# Finding *Candida auris* in public metagenomic repositories

**Jorge E. Mario-Vasquez**[1], **Ujwal R. Bagal**[2], **Elijah Lowe**[3], **Aleksandr Morgulis**[4], **John Phan**[3], **D. Joseph Sexton**[1], **Sergey Shiryev**[4], **Rytis Slatkevičius**[5], **Rory Welsh**[1], **Anastasia P. Litvintseva**[1], **Matthew Blumberg**[5], **Richa Agarwala**[4], **Nancy A. Chow**[1]*

**1** Mycotic Diseases Branch, Centers for Disease Control and Prevention, Atlanta, Georgia, United States of America, **2** ASRT Inc., Atlanta, Georgia, United States of America, **3** General Dynamics Information Technology Inc., Atlanta, Georgia, United States of America, **4** National Center for Biotechnology Information, Bethesda, Maryland, United States of America, **5** GridRepublic, Cambridge, Massachusetts, United States of America

* yln3@cdc.gov

## Abstract

*Candida auris* is a newly emerged multidrug-resistant fungus capable of causing invasive infections with high mortality. Despite intense efforts to understand how this pathogen rapidly emerged and spread worldwide, its environmental reservoirs are poorly understood. Here, we present a collaborative effort between the U.S. Centers for Disease Control and Prevention, the National Center for Biotechnology Information, and GridRepublic (a volunteer computing platform) to identify *C. auris* sequences in publicly available metagenomic datasets. We developed the MetaNISH pipeline that uses SRPRISM to align sequences to a set of reference genomes and computes a score for each reference genome. We used MetaNISH to scan ~300,000 SRA metagenomic runs from 2010 onwards and identified five datasets containing *C. auris* reads. Finally, GridRepublic has implemented a prospective *C. auris* molecular monitoring system using MetaNISH and volunteer computing.

## Introduction

*Candida auris* is an emerging and often multidrug-resistant yeast that can cause invasive candidiasis, a life-threatening disease with high mortality [1]. The World Health Organization (WHO) classified *C. auris* as a critical priority pathogen due to its high outbreak potential, resistance to most available antifungal medicines, and ability to persist in the healthcare environment despite intensive infection prevention strategies [2].

Although the pathogen was first described in Japan in 2009 [3], the earliest known *C. auris* isolates were retrospectively identified and date back to 1996 in South Korea [4]. Whole-genome sequencing (WGS) of *C. auris* isolates from four world regions revealed four phylogenetically distinct clades of this fungal pathogen wherein isolates clustered geographically (Clade I, South Asia; Clade II, East Asia; Clade III, Africa; and Clade IV, South America). This finding supported the hypothesis that *C. auris* emerged independently and simultaneously in geographically separated human populations [5]. WGS of a recently identified isolate from

**Data Availability Statement:** SRPRISM is available at https://github.com/ncbi/SRPRISM. Software for rank calculation, sequence data for reference genomes, and a sample run have been deposited at Zenodo.org and are available at https://doi.org/10.

5281/zenodo.10214980. The benchmark dataset is available at NCBI under BioProject PRJNA631031. All additional data analyzed in the manuscript is already publicly available in SRA at NCBI (https://www.ncbi.nlm.nih.gov/sra).

**Funding:** The authors received no specific funding for this work.

**Competing interests:** The authors have declared that no competing interests exist.

Iran showed the existence of a fifth major clade, with hundreds of thousands of single nucleotide polymorphisms (SNPs) separating this isolate from the four known clades [6,7]. The five major clades are separated by tens to hundreds of thousands of SNPs. Within each clade, isolates from varying countries are typically separated by hundreds to thousands of SNPs [8].

Despite intense efforts to understand how this pathogen emerged and spread to healthcare facilities worldwide, the natural reservoirs of *C. auris* are poorly understood. Two alternative hypotheses have been proposed to explain the origin of *C. auris*. One suggests that *C. auris* existed in the environment before clinical recognition and emerged as a human pathogen due to thermal adaptation in response to environmental changes [9]. Several biological properties of *C. auris*, such as thermotolerance and halotolerance, that allow this fungus to survive in hypersaline environments provide indirect evidence supporting this theory [10]. The other hypothesis is based on the *C. auris* unique propensity to colonize human skin [11,12] and suggests that *C. auris* might have existed as a minor skin commensal colonizing poorly studied sites on the human body, such as the external ear canal, in isolated human populations and emerged globally in response to the increased use of antifungals in medical and agricultural practices [10,13]. This hypothesis is indirectly supported by the results of molecular dating, which showed that the emergence of outbreak causing strains in three different lineages (Clade I, Clade III, and Clade IV) coincided with the introduction of azoles into clinics and agriculture [13]. Of course, other potential explanations are also possible, and more research is needed to better understand the environmental and human reservoirs of this pathogen. Two recent publications reported the isolation of *C. auris* from a salt marsh and sandy beach on the Andaman Islands in India and Colombia's coastal estuaries [14,15]. These findings suggest a need for further environmental and human microbiome evaluations.

To conduct extensive environmental evaluations in a financially and logistically feasible manner, investigators have utilized metagenomic data in public repositories [16] like the Sequence Read Archive (SRA), the largest global sequence repository [17]. In this study, the U. S. Centers for Disease Control and Prevention (CDC), the National Center for Biotechnology Information (NCBI), and GridRepublic partnered to develop MetaNISH (**Meta**genomic **N**eedles **I**n **S**equence **H**ay) and pipelines that utilize it. With these pipelines, we retrospectively screened ~300,000 shotgun metagenomic SRA runs from 2010 to 2022 to identify and describe datasets containing *C. auris*. In addition, we started prospectively screening datasets for this fungal pathogen daily in April 2023.

## Materials and methods

### MetaNISH design

NCBI developed the MetaNISH pipeline to screen metagenomic read sets for the presence of each genome in the given set of reference genomes (see benchmark development section). The pipeline consists of two steps: (I) the alignment step using SRPRISM [18] that aligns reads to all reference genomes as a single database, and (II) the score computation step, which increases the score for samples with reads aligned across the genome compared to samples with reads aligned to a small section of the genome.

Alignment is performed with SRPRISM as it guarantees the reporting of all equally good alignments (max 255) across all sequences in the database. Additionally, it supports specifying the region on the reads that must align and a maximum number of errors (mismatches, insertions, or deletions) in the reported alignments. For this study, we required SRPRISM to align the first 100 bases of the reads and specified a maximum of 15 errors for the reported alignments. For reads shorter than 100 bases, the full length of the read is aligned. The design of SRPRISM guarantees that the first 100 bases will have at most 5 of the 15 errors allowed. We

chose the first 100 bases as the region that must align as the read quality drops beyond that in many Illumina runs, and 100 bp is also long enough to avoid spurious matches. An example of such a read is SRR11734778.40769.1, which is a paired read with each mate of length 251 bases. Alignments for this read are included in the data released at Zenodo.org (https://doi.org/10.5281/zenodo.10214980). It was shown that the first 163 bases of the read were an exact match to *C. auris* genomes. However, the remaining portion of the read (specifically, the substring from 164 to 251) seems of inferior quality as it did not report a match to anything in the non-redundant (nr) nucleotide database at NCBI as of November 2023.

Score computation was devised for metagenomic read sets where the depth of coverage by reads could vary considerably over the genome. To determine the extent to which reads were aligned with the genome, regardless of coverage at aligned regions, *a padding of up to 100 Kb* was added to either end of the genome region aligned by the read. A scaling factor was applied to adjust the padding length for each read and genome based on the number of alignments, so this ensures that multiple mappings of a read to a genome do not exceed 100 Kb for each location. The score was the percentage of the genome covered by padded alignments. For example, read SRR11734778.40769.1 aligned to four *C. auris* genomes with only one location in each genome. Therefore, full padding of 100 Kb was used for each of the four alignments. However, read SRR11734778.1429600.1 aligned to four contigs on three genomes, as shown in Fig 1. Fig 1A and 1B show four alignments on two genomes, which reduced the padding to 25 Kb. Fig 1C shows two alignments on the third genome, which reduced the padding to 50 Kb. Padded coverage cannot extend beyond contig boundaries.

Using the reference genomes provided and empirical data from a set of 4,000 SRA runs, we proposed padding of 100 Kb and a score of at least 75 to indicate the presence of the corresponding genome in the read set. The choice is conservative, where it can potentially flag a few runs as scoring at least 75 when the genome is not present (false positives) but is unlikely to miss any (false negatives). At the same time, the parameters are not too conservative to make the false positive rate unacceptable. An example that illustrates the importance of padding regardless of the number of reads aligned at any genome location is SRR9016983. The alignment of reads from SRR9016983 to the reference genome B12037.1 is 2.8% across 1,975 locations throughout the genome. Among these reads, 50.4% have 1X coverage, and 38.7% have 2X coverage. Adding 100 Kb padding to these adjacent alignments allows the coverage to reach 100%, thus increasing the likelihood of detection.

The CPU time for the 4,000 runs used for determining the parameters varied from 25 seconds (for SRR7125652) to 17 hours 14 minutes (for SRR8550535) with a median time of 25 minutes. SRR7125652 has 86,554 paired reads, while SRR8550535 has over 418 million paired reads. We noted that SRPRISM, which takes almost all the time in the MetaNISH pipeline (as score calculation takes only a couple of seconds), can be run in multi-threaded mode with good scaling till eight threads, but we did not use that option for results reported here.

The design of MetaNISH can be used for tracking any pathogen. It requires developing a reference set with representative genomes from all clades for the pathogen to be tracked and for nearby species. Doing so allows SRPRISM to find the best matches for each read among the genomes where a match can be expected. Then, empirical analysis is needed to find suitable parameters for padding and score threshold. If read properties change substantially over time, the alignment method and parameters may need revisited.

## Benchmark development

CDC collated a set of 100 reference genomes representing priority pathogens for fungal molecular surveillance, including a representative subset of genomes for *C. auris* [19,20]. Specifically,

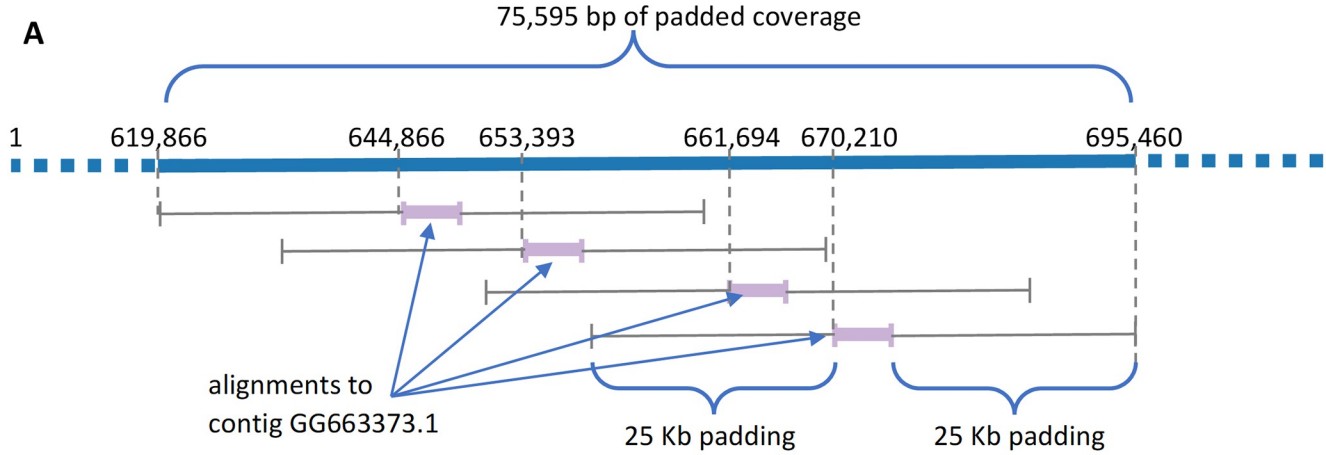

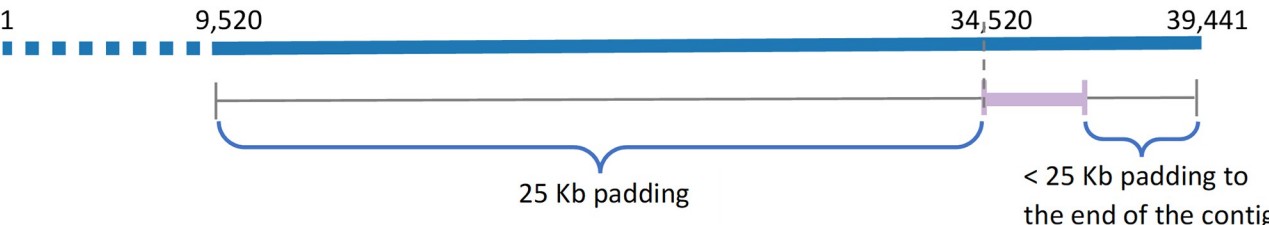

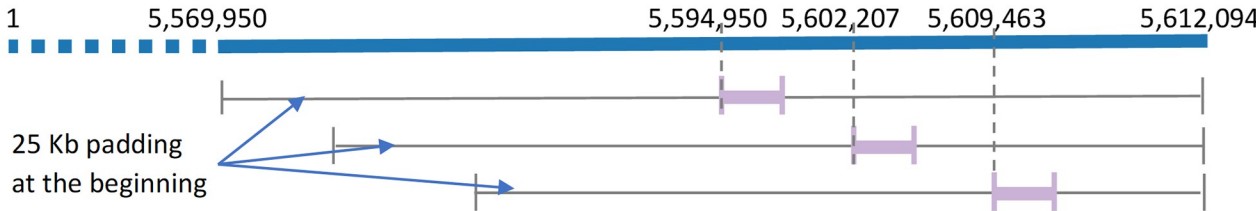

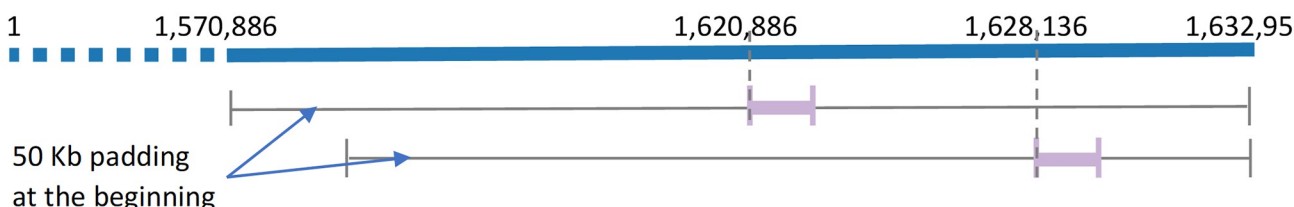

**Fig 1. Padding in alignments of read SRR11734778.1429600.1 on different assemblies.** (A) Assembly GCA_000150115.1 (B) Assembly GCA_000151005.2 (C) Assembly GCA_000151035.1.

14 genomes were of *C. auris*, 44 were of other *Candida* species, and 42 were of other fungal genera (i.e., *Ajellomyces*, *Blastomyces*, *Clavispora*, *Coccidioides*, *Cryptococcus*, *Emergomyces*, *Emmonsia*, *Paracoccidioides*, *Pichia*, *Pneumocystis*, *Saitozyma*, *Sporothrix*, and *Talaromyces*). Detailed information regarding assemblies used are found in S1 Table. The data released for this paper also includes sequences for all 100 reference genomes.

For the benchmark dataset, sequencing data for 20 metagenomic runs were generated by sequencing clinical specimens with *C. auris* spiked in at various concentrations. Briefly, residual material from *C. auris* colonization screening swabs collected from the anterior nares were used as a benchmarking dataset. The qualitative presence of *C. auris* was first confirmed by enrichment broth culture [21]. The concentration of *C. auris* cells was then assessed through a quantitative Sybr Green qPCR as previously described [22]. Cell concentrations were interpolated from a standard curve built using samples spiked with *C. auris* AR 0385 at serial dilutions ranging between $10^7$ CFU/mL to $10^3$ CFU/mL. Concentrations in the standard curve were confirmed by CFU counts and tested with three biological replicates at each concentration. The melt curve was referenced for both standard curve and benchmark samples to confirm that a strong melt peak was present in positive samples at ~83–84˚C, the signature temperature indicative of *C. auris.* No unspecific amplification was observed in the standard curve or benchmarking samples. Five "no template controls" were included in the run. As expected, there was no amplification in these samples. Sequence data were deposited in PRJNA631031.

## MetaNISH implementation

The bash pipeline used by CDC for searching NCBI's SRA database integrated with MetaNISH is depicted in Fig 2. Following filtering criteria were applied (*platform*: Illumina; *library_source*: metagenomic; *consent*: public; *assay_type*: WGS; *library_selection*: random) to download only whole-genome sequence metagenomic datasets temporarily with additional metadata (accession ID, biosample, bioproject, release date, library layout, mbases, and organism). The alignment and scoring were done as per the MetaNISH design described earlier. The scores for all 100 reference genomes for each SRA ID are reported by the pipeline.

## Data analysis

The samples in the benchmark set were spiked using *C. auris* AR 0385 (Biosample SAMN05379620 as per CDC's AR isolate bank; strain B11244) that has reads in SRA under accession SRR3883465 but no published assembly. Therefore, we used the assembly in our reference set of 100 genomes that is closest to the spike in strain for presenting the data analysis. We found the closest assembly in the following manner: The reads in SRR3883465 were assembled using SKESA [23], resulting in an assembly with a length of 12.21 Mb and N50 of 22 Kb. The assembly was then aligned to all reference genomes using BLAST, retaining only the best e-value alignments, and coverage on the reference genomes was determined using the retained alignments. The analysis revealed that 12.19 Mb of the assembly had alignments to reference genomes, with the maximum coverage for the reference assembly B12342.1 at 11.52 Mb aligned. The second-best coverage was for B11245, but it had only 1.4 Mb aligned. All alignments to B12342.1 had a percent identity of at least 99.6%, of which all except 8,465 bp aligned at a percent identity of at least 99.9%. Hence, MetaNISH scores for the benchmark dataset were presented using reference genome B12342.1. These scores were compared to KrakenUniq [24], a method for metagenomic classification that provides a quantitative measure of genome coverage. KrakenUniq was run with defaults, except no information was printed for unclassified sequences using parameter—only-classified-output.

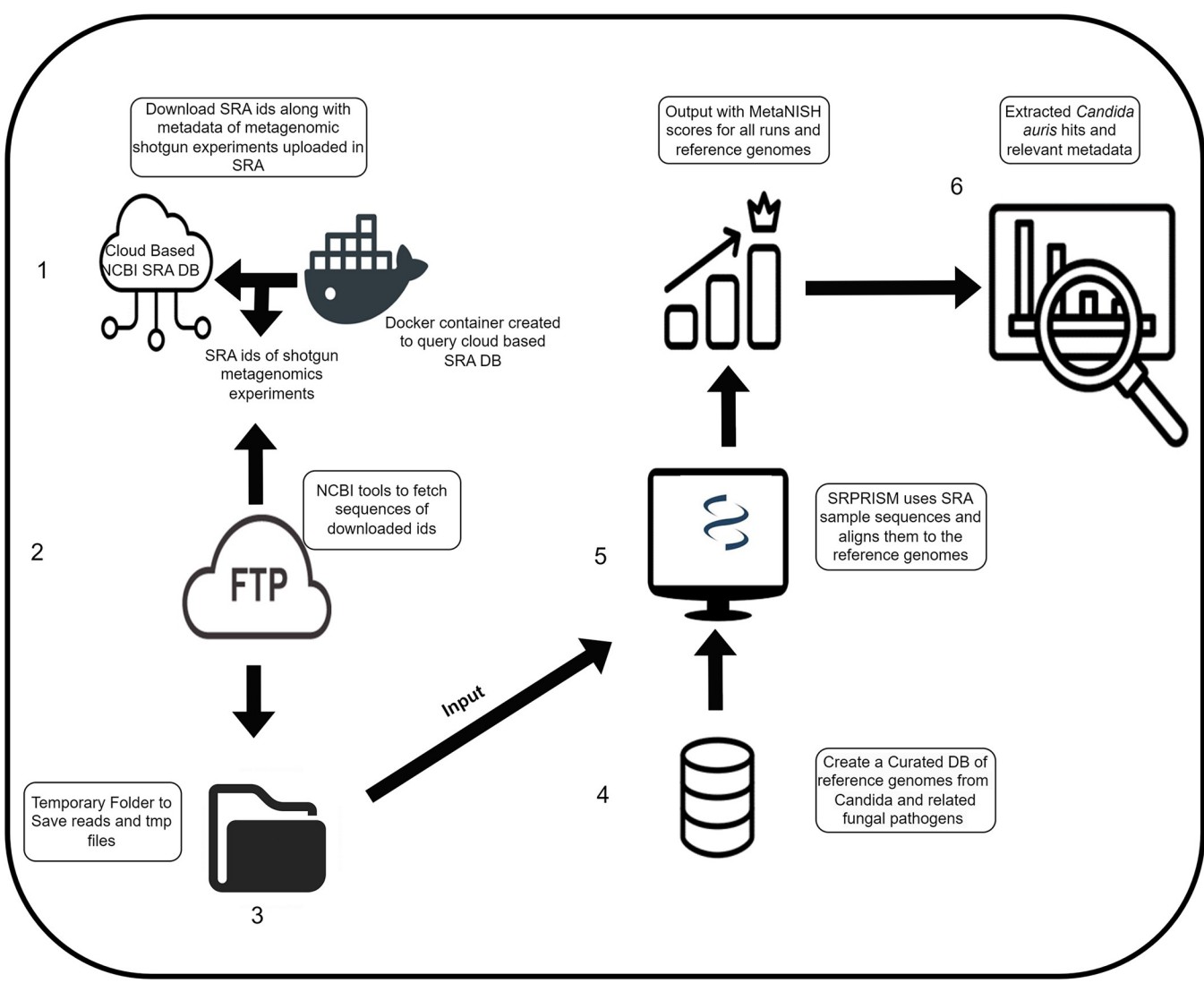

**Fig 2. Pipeline for *C. auris* sequence-based monitoring using MetaNISH.** Steps 1–4 comprise collecting the required input data (samples sequence reads and reference database) for MetaNISH (step 5), whose output is a file with the scores for all references for each sample processed. Finally (step 6), this stack of files is processed and analyzed to obtain the samples with positive hits (score ≥ 75) of *C. auris*.

Heatmaps were generated using alignments to reference genome B12342.1, contigs in the reference assembly were split into consecutive intervals of size 200 Kb and 2 Kb to represent padding of 100 Kb and 1 Kb on both ends of the alignments for reads, respectively. For each alignment, the starting position of the alignment on the contig was used to determine the bin where the alignment contributes to the count and to increase the count for that bin by one. The counts were plotted in MATLAB (version R2020a, Update 2) using the *imagesc* function to produce the heatmaps.

SRA reads were aligned to the set of reference genomes, and a score for each reference genome using padded coverage was obtained for SRA ids from January 2010 to November 2022, retrospectively. Using the output from MetaNISH, we scanned the scores for all *C. auris* reference assemblies using a MetaNISH score ≥50 up to the maximum possible score of 100 to obtain the number of SRA runs with at least a hit on any of the *C. auris* assemblies. With the

suggested score of $\geq 75$ as the threshold for positive pathogen identification, samples with *C. auris* positive hits were described using the metadata collected.

## Results

### Benchmarking the reference dataset in the monitoring tool

The benchmark dataset is further described in Table 1. The presence of *C. auris* spike-in, cell concentration, scores computed by MetaNISH using different padding lengths, and the assembly coverage reported by KrakenUniq (Table 1 and Fig 3) are indicated for each metagenomic run. Using a score of 75 with 100 Kb padding, MetaNISH was able to detect all true positive as well as one false positive sample, while a score of 80 was able to separate all positive and negative samples. Significant variation was observed in scores with padding of less than 100 Kb, no padding, and KrakenUniq. For example, SRR11734778 compared to SRR11734781 had similar cell concentrations ($7.1 \times 10^4$ CFU/mL and $6.7 \times 10^4$ CFU/mL, respectively) and the same score (100) using 100 Kb padding; however, SRR11734778 compared to SRR11734781 had substantially different scores with 1 Kb padding (39.97 and 89.09, respectively), no padding (7.42 and 24.38, respectively) and KrakenUniq (3.73 and 13.52, respectively). Increasing padding to even just 10 Kb brings the padded coverage to over 98 for SRR11734778 and SRR11734781. As reflected in Table 1 and Fig 3, a padding length of 100 Kb was found to be effective in differentiating positive samples like SRR11734782 with a score of 92.74 from negative samples, while MetaNISH, with lesser than 100 Kb or no padding and KrakenUniq coverage values were not effective. Fig 3 depicted that the score is not affected by the number of

**Table 1. Benchmark results using B12342 reference genome.**

| Benchmark design | | | MetaNISH scores with the specified padding | | | | | | KrakenUniq |
|---|---|---|---|---|---|---|---|---|---|
| Run | Status | Concentration [a] | 150 Kb | 100 Kb | 50 Kb | 10 Kb | 1 Kb | None | coverage*100 |
| SRR11734785 | pos | $5.8 \times 10^5$ | 100 | 100 | 100 | 100 | 99.98 | 99.9 | 82.9 |
| SRR11734772 | pos | $3.5 \times 10^5$ | 100 | 100 | 100 | 100 | 99.97 | 99.57 | 76.98 |
| SRR11734791 | pos | $2.0 \times 10^5$ | 100 | 100 | 100 | 100 | 99.96 | 85.92 | 56.38 |
| SRR11734780 | pos | $1.6 \times 10^5$ | 100 | 100 | 100 | 100 | 99.93 | 97.88 | 71.1 |
| SRR11734775 | pos | $1.2 \times 10^5$ | 100 | 100 | 100 | 100 | 99.32 | 50.33 | 30.6 |
| SRR11734777 | pos | $8.6 \times 10^4$ | 100 | 100 | 100 | 99.98 | 89.8 | 30.07 | 17.36 |
| SRR11734778 | pos | $7.1 \times 10^4$ | 100 | 100 | 100 | 98.8 | 39.97 | 7.42 | 3.73 |
| SRR11734781 | pos | $6.7 \times 10^4$ | 100 | 100 | 100 | 100 | 89.09 | 24.38 | 13.52 |
| SRR11734776 | pos | $4.3 \times 10^4$ | 100 | 100 | 100 | 99.98 | 66.17 | 13.89 | 9.003 |
| SRR11734779 | pos | $2.7 \times 10^4$ | 100 | 100 | 100 | 99.99 | 68.22 | 13.6 | 7.575 |
| SRR11734773 | pos | $1.1 \times 10^4$ | 100 | 100 | 100 | 99.99 | 93.81 | 29.67 | 16.4 |
| SRR11734774 | pos | $1.1 \times 10^4$ | 100 | 100 | 99.8 | 71.75 | 14.28 | 2.36 | 1.756 |
| SRR11734783 | pos | $1.0 \times 10^4$ | 100 | 100 | 100 | 99.74 | 48.4 | 8 | 4.61 |
| SRR11734784 | pos | $2.9 \times 10^3$ | 100 | 100 | 100 | 99.99 | 67.56 | 13.65 | 8.118 |
| SRR11734782 | pos | $1.9 \times 10^3$ | 96.72 | 92.74 | 74.01 | 23.71 | 3.04 | 0.43 | 0.8356 |
| SRR11734790 | neg | NA | 85.08 | 79.43 | 53.74 | 14.74 | 1.84 | 0.25 | 0.9267 |
| SRR11734789 | neg | NA | 65.16 | 57.18 | 34.14 | 8.43 | 0.99 | 0.13 | 0.4599 |
| SRR11734787 | neg | NA | 50.28 | 41.88 | 22.96 | 5.29 | 0.68 | 0.11 | 0.1616 |
| SRR11734788 | neg | NA | 46.98 | 39.93 | 22.75 | 5.4 | 0.63 | 0.08 | 0.8494 |
| SRR11734786 | mock | NA | 3.56 | 2.88 | 1.47 | 0.57 | 0.14 | 0.04 | 0.4101 |

**Pos**: Positive for *C. auris*; **neg**: Negative for *C. auris*; **mock**: Pooled skin swab samples negative for *C. auris*.

[a] Units in CFU/mL.

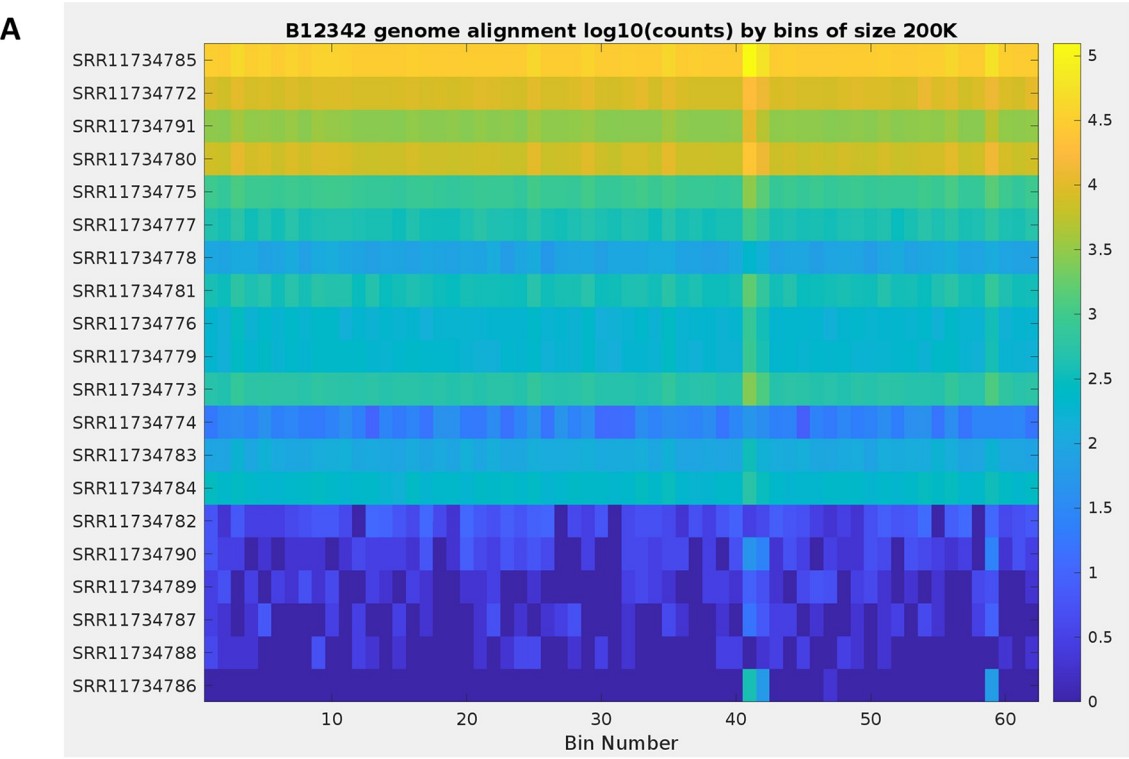

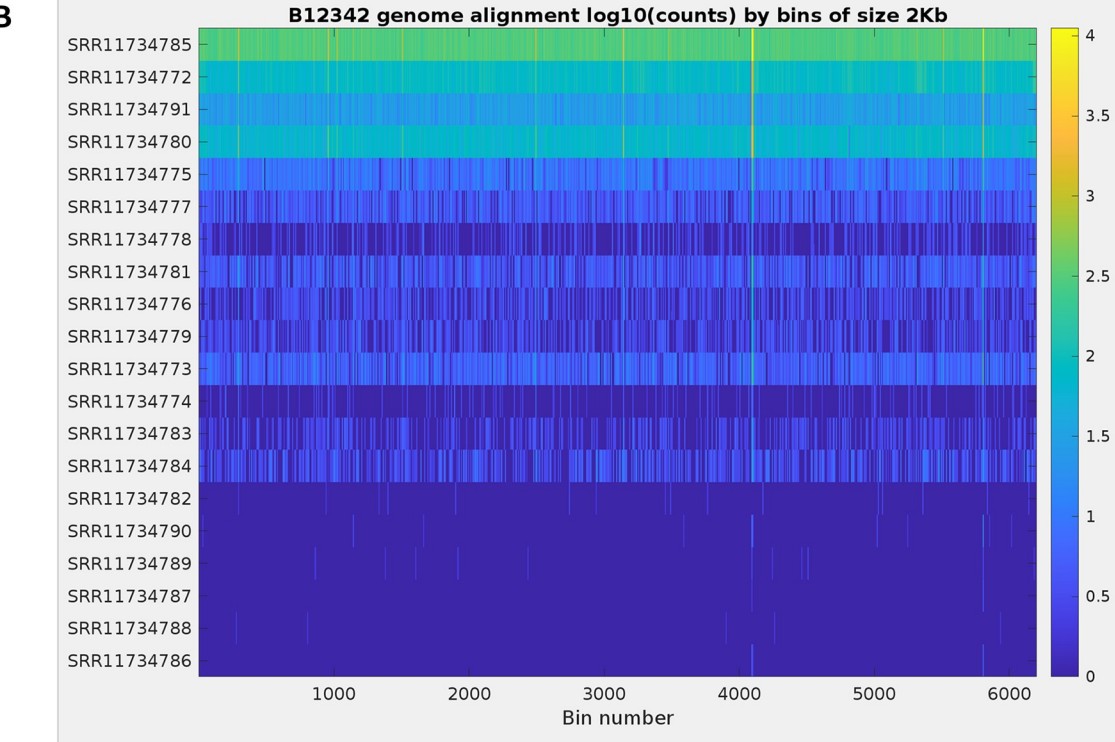

**Fig 3. Distribution of the number of reads aligned in the benchmark set to the B12342 reference genome.** The reference genome is binned by 200 Kb to reflect padding of 100 Kb on both ends of read alignments (A) and similarly by 2 Kb to reflect the padding of 1 Kb (B). The read sets are sorted by decreasing concentration levels of *C. auris*, with the topmost run SRR11734785 having the highest concentration.

**Table 2. Number of SRA records scanned.**

| Year | Records Scanned | Cumulative |
|------|----------------|-----------|
| **2010** | 1 | 1 |
| **2011** | 41 | 42 |
| **2012** | 551 | 593 |
| **2013** | 2570 | 3163 |
| **2014** | 7023 | 10186 |
| **2015** | 17402 | 27588 |
| **2016** | 11529 | 39117 |
| **2017** | 21458 | 60575 |
| **2018** | 26897 | 87472 |
| **2019** | 51395 | 138867 |
| **2020** | 47766 | 186633 |
| **2021** | 46952 | 233585 |
| **2022** | 57756 | 291341 |

reads aligning in a specific bin, such as high coverage for SRR11734785 (all yellow) and relatively low but well distributed throughout the genome coverage for SRR11734774 (primarily light blue).

## Detection of *Candida auris* in metagenomic datasets

Using MetaNISH (Fig 2), the number of samples per year that met the filtering criteria increased from one sample in 2010 to 57,756 in 2022 (Table 2). As of December 2022, 291,341 SRA samples were analyzed (Table 2) to produce an output of *C. auris* hits with varying genome coverage (Table 3). *C. auris* was identified in five sample datasets: PRJNA488992 (2 SRA runs), PRJNA657014 (4 SRA runs), PRJNA475330 (1 SRA run), PRJNA631031 (15 SRA

**Table 3. Binned padded genome coverage (score) of SRA runs (samples) with *Candida auris* hits.**

| Year | Score Ranges | | | | |
|------|--------|--------|--------|---------|------|
| | [50–75) | [75–85) | [85–95) | [95–100) | 100* |
| **2010** | 0 | 0 | 0 | 0 | 0 |
| **2011** | 0 | 0 | 0 | 0 | 0 |
| **2012** | 0 | 0 | 0 | 0 | 0 |
| **2013** | 0 | 0 | 0 | 0 | 0 |
| **2014** | 0 | 0 | 0 | 0 | 0 |
| **2015** | 0 | 0 | 0 | 0 | 0 |
| **2016** | 5 | 0 | 0 | 0 | 0 |
| **2017** | 0 | 0 | 0 | 0 | 0 |
| **2018** | 0 | 0 | 0 | 0 | 0 |
| **2019** | 1 | 0 | **2** | **4** | **6** |
| **2020** | 6 | **3** | **5** | **9** | **14** |
| **2021** | 0 | 0 | 0 | 0 | 0 |
| **2022** | 1 | 0 | 0 | 0 | 0 |
| **Total** | 13 | **3** | **7** | **13** | **20** |

Numbers in bold correspond to the samples where *C. auris* was identified, and its metadata is shown in Table 4.

*This is not an interval; it equals the number of runs with a score of 100. For all other columns, the interval is closed at the beginning and open at the end.

**Table 4. Bioproject metadata for samples with WGS data at SRA with *C. auris* positive hits.**

| Run Record | Score | Release Year | Bioproject | SRA study | Title | Environment or isolation source |
|---|---|---|---|---|---|---|
| SRR8584355 | 100% | 2019 | PRJNA488992 | SRP159446 | Metagenomics of wastewater drains and river samples from Delhi, India | Wastewater drain |
| SRR8584356 | 100% | | | | | Urban river |
| SRR9016982 | 100% | 2019 | PRJNA657014 | SRP277451 | Sequencing data from point prevalence study associated with *C. auris* Raw sequence reads | Combined axilla and inguinal crease (groin) and anterior nares (Human skin metagenome) |
| SRR9016983 | 100% | | | | | |
| SRR9016984 | 100% | | | | | |
| SRR9016985 | 100% | | | | | |
| SRR10237756 | >90% | 2019 | PRJNA475330 | SRP161559 | Metagenomic assembly of the iron-reducing, 1-methylnaphthalene-degrading enrichment culture (1MN) | Sulfur-oxidizing nitrate-reducing enrichment culture |
| SRR11734772 | 100% | 2020 | PRJNA631031 | SRP260772 | Study of microbial diversity of anterior nares swabs from patients colonized by the pathogen *Candida auris* | Human nasopharyngeal metagenome |
| SRR11734773 | 100% | | | | | |
| SRR11734774 | 100% | | | | | |
| SRR11734775 | 100% | | | | | |
| SRR11734776 | 100% | | | | | |
| SRR11734777 | 100% | | | | | |
| SRR11734778 | 100% | | | | | |
| SRR11734779 | 100% | | | | | |
| SRR11734780 | 100% | | | | | |
| SRR11734781 | 100% | | | | | |
| SRR11734783 | 100% | | | | | |
| SRR11734784 | 100% | | | | | |
| SRR11734785 | 100% | | | | | |
| SRR11734791 | 100% | | | | | |
| SRR11734782 | >90% | | | | | |
| SRR10680803 | >90% | 2020 | PRJNA557323 | SRP237407 | Human gut metagenomes from Hong Kong populations | Stool samples (Human gut metagenome) |
| SRR10680804 | | | | | | |

runs), and PRJNA557323 with two SRA runs (Table 4). Sequence reads from PRJNA488992 were collected from wastewater drains and river samples from Delhi, India. Reads from PRJNA657014 and PRJNA631031 datasets were from skin swabs of the residents of healthcare facilities in the United States where *C. auris* had been identified [25] and our benchmark dataset, respectively. Sequence reads from PRJNA557323 were collected from human stool samples from Hong Kong. Finally, PRJNA475330 samples were collected from Germany's sulfur-oxidizing nitrate-reducing enrichment culture of a groundwater sample (Table 4).

### Prospective monitoring

GridRepublic has implemented a molecular monitoring system using MetaNISH on a volunteer computing platform (i.e., a distributed computing platform comprised of resources volunteered by the general public). This system successfully screens all new metagenomic data submitted to SRA daily for *C. auris* (averaging 925 runs per day). These results are available on the web at www.gridrepublic.org/biosurveillance.

### Discussion

In a collaborative effort between CDC, NCBI, and GridRepublic, we developed and benchmarked bioinformatics tools for the prospective monitoring of metagenomic datasets for the detection of *C. auris* and examined ~300,000 SRA runs released between 2010 and 2022 to

identify this pathogen. Using benchmarking samples generated by spiking human skin microbiome with known concentrations of *C. auris*, we found that the MetaNISH pipeline with the following parameters was successful in identifying samples with *C. auris*: (i) alignment of the first 100 bases to the target using SRPRISM, (ii) padding length of 100 Kb, and (iii) score threshold of at least 75. Increasing a cutoff score to 80 was able to separate all positive and negative samples; however, using a cutoff of 75 may increase the chances of identifying samples with a low prevalence of *C. auris*. The proposed parameters were especially beneficial for the detection of benchmarking samples with lower concentrations of *C. auris* reads, such as SRR11734774 and SRR11734782, which showed low base pair coverage of 2.36% and 0.43% by SRPRISM, and 1.8% and 0.8%, by KrakenUniq but generated scores of 100 and 92.4 with MetaNISH and 100 Kb padding (Table 1). The alignment scores above 90 indicate that the alignments were well-distributed throughout the genome, increasing confidence in the results.

Our study identified five metagenomic datasets containing *C. auris* sequences in the public SRA repository. The first was the benchmark dataset used to test our pipeline. The second was from skin swabs of patients colonized with *C. auris* (24). Detection of *C. auris* in these samples was not surprising, although important for providing an independent validation of the developed method. The third dataset was from a study of stool microbiome of healthy individual in Hong Kong [26], which was novel and unexpected. Although *C. auris* has previously been isolated from the gastrointestinal tract [27,28], it is generally accepted that this fungus is primarily a skin colonizer [12]. Several publications show that *Candida* spp. can survive to passage through the gut in healthy adults and possibly generate further spread via wastewater [29,30]. Our observation raises the question of whether patients colonized with *C. auris* on the skin are also colonized in the gut and whether some human communities may harbor the previously unknown reservoirs of *C. auris* [31,32]. More studies of healthy people are needed to understand the prevalence of *C. auris* in the community [33].

The presence of *C. auris* in the fourth set, laboratory culture of the iron-reducing bacteria most likely indicates contamination [34], although it suggests its ability to survive under such iron-reducing conditions. The detection of *C. auris* in aquatic biome samples from Delhi is consistent with the recent report showing isolation of *C. auris* from the coastal waters in India and Colombia [14,15], which provided support to the hypothesis of an environmental origin of this pathogen [9,10]. However, it is also equally likely that *C. auris* might have been spread into aquatic environment from contaminated wastewater after being excreted from the gastrointestinal tract or washed off the skin of a colonized people [35]. These findings point to its most likely mode of spread (any aquatic stream or aqueous medium) between the human populations and environment and vice versa [36].

A limitation of this study is that only shotgun metagenomic data were analyzed, and amplicon sequencing data were excluded. Relatively high costs and the need for more advanced bioinformatics have limited the use of shotgun metagenomics for microbiome analysis on a large scale [37–39]. In contrast, the amplicon sequencing approach is the most widely used method for analyzing microbial communities due to its cost-effectiveness, established data analysis pipelines, and availability of an extensive archive of reference data [12,25,36]. Thus, building and validating a search option for amplicon datasets into MetaNISH by generating benchmark amplicon datasets and *C. auris* reference databases will complement the existing metagenomic search function. The other limitation of the study is that for most SRA submissions, only limited metadata on the specimens is available. A follow-up with the submitters is often needed to identify additional details of the study and to determine whether a finding of *C. auris* sequences in the sample is indeed a reflection of the sequenced community and not an artifact of laboratory practices, in which *C. auris* might have been used as a loading control or occurred as a contaminant. It is also important to point out that the identification of reads of

*C. auris* in certain samples may not necessarily indicate that these samples represent the ecological niche for this fungus. As described above with an example of finding *C. auris* in coastal waters, the directionality of *C. auris* transition between the human skin and coastal waters is not clear. It is equally likely that the fungus might have emerged in costal habitats and later transitioned into human population, or in contrast, that it has emerged elsewhere and was introduced into the coastal waters from colonized persons.

Because, in many cases, there is a significant time lag between the collection of a sample and the submission/publication of its sequence reads, the detections that can be made (even daily) do not imply an active outbreak response but are valuable *post hoc* information that allows tracking trends of spread and are encompassed in the data integration of a One Health surveillance system [40].

The findings presented in this study using MetaNISH on public metagenomic data support the results of previous work on *C. auris* in natural environments. This work also lays the foundation for the prospective monitoring system for *C. auris* because the modular design of MetaNISH makes it suitable for this daily job, which in addition to addressing scientific questions about the origin of *C. auris*, provides a necessary public health monitoring tool for investigating the spread of *C. auris* into the new areas. GridRepublic has implemented the pipeline developed and evaluated in our study on a distributed computing network, adapting it into a real-time monitoring system. Future efforts can adapt this tool to monitor other emerging pathogens and public health threats.

## Supporting information

**S1 Table. Reference genomes.**
(DOCX)

## Acknowledgments

This research work was supported in part by the National Center for Biotechnology Information of the National Library of Medicine (NLM), National Institutes of Health. This work was also made possible through support from the CDC Office of Advanced Molecular Detection (OAMD). The contents of this publication are solely the responsibility of the authors and do not necessarily represent the official views of the Centers for Disease Control and Prevention.

## Author Contributions

**Conceptualization:** Richa Agarwala, Nancy A. Chow.

**Data curation:** Jorge E. Mario-Vasquez, Ujwal R. Bagal, Richa Agarwala.

**Formal analysis:** Jorge E. Mario-Vasquez, Ujwal R. Bagal, Richa Agarwala.

**Funding acquisition:** Rory Welsh, Nancy A. Chow.

**Investigation:** Jorge E. Mario-Vasquez, Ujwal R. Bagal, Richa Agarwala.

**Methodology:** Jorge E. Mario-Vasquez, Ujwal R. Bagal, Elijah Lowe, Aleksandr Morgulis, John Phan, Sergey Shiryev, Rytis Slatkevičius, Rory Welsh, Matthew Blumberg, Richa Agarwala.

**Project administration:** Nancy A. Chow.

**Resources:** D. Joseph Sexton, Matthew Blumberg, Richa Agarwala, Nancy A. Chow.

**Software:** Jorge E. Mario-Vasquez, Ujwal R. Bagal, Elijah Lowe, Aleksandr Morgulis, John Phan, Sergey Shiryev, Rytis Slatkevičius, Matthew Blumberg, Richa Agarwala.

**Supervision:** Richa Agarwala, Nancy A. Chow.

**Validation:** Jorge E. Mario-Vasquez, Ujwal R. Bagal, Richa Agarwala.

**Visualization:** Jorge E. Mario-Vasquez, Ujwal R. Bagal, Richa Agarwala.

**Writing – original draft:** Jorge E. Mario-Vasquez.

**Writing – review & editing:** Jorge E. Mario-Vasquez, Ujwal R. Bagal, D. Joseph Sexton, Rory Welsh, Anastasia P. Litvintseva, Matthew Blumberg, Richa Agarwala, Nancy A. Chow.

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
