## [Decision Letter · Decision Letter 0]

31 Oct 2023

PONE-D-23-27761Finding Candida auris in public metagenomic repositoriesPLOS ONE

Dear Dr. Chow,

Thank you for submitting your manuscript to PLOS ONE. After careful consideration, we feel that it has merit but does not fully meet PLOS ONE’s publication criteria as it currently stands. Therefore, we invite you to submit a revised version of the manuscript that addresses the points raised during the review process.

We look forward to receiving your revised manuscript.

Kind regards,

Ricardo Santos

Academic Editor

PLOS ONE

Journal Requirements:

**Additional Editor Comments:**

Please see below the comments and suggested MAJOR revisions made by the individual(s) who reviewed your manuscript.  If provided, the referee's report(s) indicate the revisions that need to be made before it can be accepted for publication.

Reviewers' comments:

Reviewer's Responses to Questions

**Comments to the Author**

1. Is the manuscript technically sound, and do the data support the conclusions?

Reviewer #1: Partly

Reviewer #2: Yes

Reviewer #3: Yes

2. Has the statistical analysis been performed appropriately and rigorously? 

Reviewer #1: Yes

Reviewer #2: N/A

Reviewer #3: Yes

3. Have the authors made all data underlying the findings in their manuscript fully available?

Reviewer #1: Yes

Reviewer #2: Yes

Reviewer #3: Yes

4. Is the manuscript presented in an intelligible fashion and written in standard English?

Reviewer #1: Yes

Reviewer #2: Yes

Reviewer #3: Yes

5. Review Comments to the Author

Reviewer #1: The manuscript presented by Mario-Vasquez et al. describing the finding of C. auris in WGS data using “MetaNISH” is very well written and can be divided into two main parts: i) the presentation of a tool that can identify C. auris in metagenomics sequencing data; and ii) the screening of public repositories for the presence of this species in metagenomics data. For this reason, it is of interest for a broad community. Still, I have some comments/concerns that I would like to clarify:

1. The manuscript describes the MetaNISH tool and states its availability through an NCBI ftp address. This is not the most standard way to comply with the FAIR principles. It would be good to have the code of the tool freely available in an open-source repository, such as github or gitlab.

2. Regarding the behavior of MetaNISH, it was not clear to me why only the first 100bp of a read are aligned to the reference database.

3. The padding strategy applied by the authors is very interesting and seems to be of extreme relevance for the quality of the results. How was the 100kb padding determined as the best threshold? In the manuscript I see a comparison between 1kb and 100kb, which is a huge difference. Were intermediate distances also tested?

4. The authors describe a “collage” of reference genomes, which I assume correspond to the MetaNISH reference database. It would be important to know which genomes correspond to this reference and the criteria used to choose them. Moreover, the authors mention 44 other Candida species, and 42 other fungal genera, but which species and assemblies? A supplementary table with this information would be important for transparency.

5. Still regarding the reference database, it is not clear to me whether i) the reads are aligned to the combination of all these species, or ii) exclusively to the assemblies of the species of interest. If i), how does MetaNISH deals with multimappings and how does this affect the score? If ii), by accepting a 5% error rate, how does MetaNISH ensures that the reads correspond to C. auris and not to a close-related species? It is important to clarify this in the text.

6. In Figure 1, the authors describe the pipeline used for the whole analysis and MetaNISH is just a small part of it. It would be important to clarify in the text and in the caption of the figure what were the pre-MetaNISH steps. It is not clear to me, for example, why the Docker logo is used without any reference to it in the manuscript. Moreover, as the “Curated Reference Database” is outside the MetaNISH, does this mean that MetaNISH can be run with any reference database created by the user? I think the manuscript would benefit if this Figure was substantially improved, clarifying not only the pre-MetaNISH steps, but also, and more importantly, the MetaNISH workflow and its input/output files.

7. Regarding the benchmarking and comparison with KrakenUniq, what were the settings used for KrakenUniq? Also, how much time does MetaNISH require for the analysis of an SRA entry?

8. Sample SRR11734778 seems to have an outlier behavior when compared to the others (Table 1). Do the authors have an idea of what can be influencing the score in this sample?

9. Table 3 needs more explanation. Does “score” correspond to “genome coverage”? And the numbers correspond to the number of samples? Why are some numbers in red?

10. Regarding the MetaNISH implementation in GridRepublic, does it specifically screen for C. auris? Or for all the species present in the reference database?

Thank you.

Reviewer #2: The authors describe an interesting system for searching for Candida auris using metagenomic data deposited in the NCBI public database. They present their analyses clearly and in detail, and also explain the many limitations of their method. This method looks very promising not only for C. auris, but also for searching for other fungal pathogens in the environment, determining the microbiome and proposing possible hypotheses for the emergence of C. auris.

One question; do you think these results can be modified if you used other strain from different clades to determine the parameters?

Reviewer #3: I find the manuscript very interesting and thorough, and the method they propose appears very sensitive in detecting C.auris infections. I have no major comments about the paper.

Minor comment: I find the description of "padding" confusing, and particularly this: "For each read and each genome, the padding length was scaled down by the number of alignments to the genome by that read. The score was the percentage of the genome covered by padded alignments." I suggest using a formula to make it clearer and more motivated, and possibly a figure to illustrate what the padding achieves.

6. PLOS authors have the option to publish the peer review history of their article (what does this mean?). If published, this will include your full peer review and any attached files.

Reviewer #1: No

Reviewer #2: No

Reviewer #3: **Yes: **Rahul Siddharthan

---

## [Author Response · Author response to Decision Letter 0]

15 Dec 2023

Review Comments to the Author

Reviewer #1: The manuscript presented by Mario-Vasquez et al. describing the finding of C. auris in WGS data using “MetaNISH” is very well written and can be divided into two main parts: i) the presentation of a tool that can identify C. auris in metagenomics sequencing data; and ii) the screening of public repositories for the presence of this species in metagenomics data. For this reason, it is of interest for a broad community. Still, I have some comments/concerns that I would like to clarify:

1. The manuscript describes the MetaNISH tool and states its availability through an NCBI ftp address. This is not the most standard way to comply with the FAIR principles. It would be good to have the code of the tool freely available in an open-source repository, such as github or gitlab.

MetaNISH uses SRPRISM that is already published and available on GitHub at https://github.com/ncbi/SRPRISM.

A snapshot of the code for computing score, sequences for reference genomes, and supporting data was created on Zenodo (https://doi.org/10.5281/zenodo.10214980), which is more appropriate for creating a publicly available snapshot of such data.

2. Regarding the behavior of MetaNISH, it was not clear to me why only the first 100bp of a read are aligned to the reference database.

In section MetaNISH Design, Materials and methods, lines 83-92, now reads:

For reads shorter than 100 bases, the full length of the read is aligned. The design of SRPRISM guarantees that the first 100 bases will have at most 5 of the 15 errors allowed. We chose the first 100 bases as the region that must align as the read quality drops beyond that in many Illumina runs, and 100 bp is also long enough to avoid spurious matches. An example of such a read is SRR11734778.40769.1, which is a paired read with each mate of length 251 bases. Alignments for this read are included in the data released at Zenodo.org (https://doi.org/10.5281/zenodo.10214980). It was shown that the first 163 bases of the read were an exact match to C. auris genomes. However, the remaining portion of the read (specifically, the substring from 164 to 251) seems of inferior quality as it did not report a match to anything in the non-redundant (nr) nucleotide database at NCBI as of November 2023.

3. The padding strategy applied by the authors is very interesting and seems to be of extreme relevance for the quality of the results. How was the 100kb padding determined as the best threshold? In the manuscript I see a comparison between 1kb and 100kb, which is a huge difference. Were intermediate distances also tested?

In section MetaNISH Design, Materials and methods, lines 109-113, now reads:

Using the reference genomes provided and empirical data from a set of 4,000 SRA runs, we proposed padding of 100 Kb and a score of at least 75 to indicate the presence of the corresponding genome in the read set. The choice is conservative, where it can potentially flag a few runs as scoring at least 75 when the genome is not present (false positives) but is unlikely to miss any (false negatives). At the same time, the parameters are not too conservative to make the false positive rate unacceptable.

In the section Benchmarking the reference dataset in the monitoring tool, Results, lines 211-214, now reads:

As reflected in Table 1 and Figure 3, a padding length of 100 Kb was found to be effective in differentiating positive samples like SRR11734782 with a score of 92.74 from negative samples, while MetaNISH, with lesser than 100 Kb or no padding and KrakenUniq coverage values were not effective.

Table 1 (lines 222-225) has been updated, indicating several padding lengths tested:

Table 1. Benchmark results using B12342 reference genome.

Benchmark design MetaNISH scores with the specified padding KrakenUniq

Run Status Concentration a 150 Kb 100 Kb 50 Kb 10 Kb 1 Kb None coverage*100

SRR11734785 pos 5.8 x 105 100 100 100 100 99.98 99.9 82.9

SRR11734772 pos 3.5 x 105 100 100 100 100 99.97 99.57 76.98

SRR11734791 pos 2.0 x 105 100 100 100 100 99.96 85.92 56.38

SRR11734780 pos 1.6 x 105 100 100 100 100 99.93 97.88 71.1

SRR11734775 pos 1.2 x 105 100 100 100 100 99.32 50.33 30.6

SRR11734777 pos 8.6 x 104 100 100 100 99.98 89.8 30.07 17.36

SRR11734778 pos 7.1 x 104 100 100 100 98.8 39.97 7.42 3.73

SRR11734781 pos 6.7 x 104 100 100 100 100 89.09 24.38 13.52

SRR11734776 pos 4.3 x 104 100 100 100 99.98 66.17 13.89 9.003

SRR11734779 pos 2.7 x 104 100 100 100 99.99 68.22 13.6 7.575

SRR11734773 pos 1.1 x 104 100 100 100 99.99 93.81 29.67 16.4

SRR11734774 pos 1.1 x 104 100 100 99.8 71.75 14.28 2.36 1.756

SRR11734783 pos 1.0 x 104 100 100 100 99.74 48.4 8 4.61

SRR11734784 pos 2.9 x 103 100 100 100 99.99 67.56 13.65 8.118

SRR11734782 pos 1.9 x 103 96.72 92.74 74.01 23.71 3.04 0.43 0.8356

SRR11734790 neg NA 85.08 79.43 53.74 14.74 1.84 0.25 0.9267

SRR11734789 neg NA 65.16 57.18 34.14 8.43 0.99 0.13 0.4599

SRR11734787 neg NA 50.28 41.88 22.96 5.29 0.68 0.11 0.1616

SRR11734788 neg NA 46.98 39.93 22.75 5.4 0.63 0.08 0.8494

SRR11734786 mock NA 3.56 2.88 1.47 0.57 0.14 0.04 0.4101

Pos: positive for C. auris; neg: negative for C. auris; mock: pooled skin swab samples negative for C. auris. a Units in CFU/mL.

4. The authors describe a “collage” of reference genomes, which I assume correspond to the MetaNISH reference database. It would be important to know which genomes correspond to this reference and the criteria used to choose them. Moreover, the authors mention 44 other Candida species, and 42 other fungal genera, but which species and assemblies? A supplementary table with this information would be important for transparency.

Following the reviewer’s suggestion, the supplementary table S1 was created. It contains information about the assemblies used in our reference database. 

In addition to the comprehensive list of C. auris genomes in which we tried to cover the diversity of clades known up to that time. We included references mainly to pathogenic fungi of public health interest, which, given the knowledge at that time, we would not necessarily expect to find in environments other than hospital environments and, like C. auris, to see if they could have other sources of origin/dispersion.

Additional text has been added at lines 137-139 of the Benchmark Development section, Materials and methods, relating Table S1 to the main manuscript.

5. Still regarding the reference database, it is not clear to me whether i) the reads are aligned to the combination of all these species, or ii) exclusively to the assemblies of the species of interest. If i), how does MetaNISH deals with multimappings and how does this affect the score? If ii), by accepting a 5% error rate, how does MetaNISH ensures that the reads correspond to C. auris and not to a close-related species? It is important to clarify this in the text.

Reads are aligned to each assembly on the reference database, and then the alignments for each SRA run are ranked according to the percentage of padded coverage. This is stated in the manuscript as follows:

- Lines 75- 78: The pipeline consists of two steps: (I) the alignment step using SRPRISM (18) that aligns reads to all reference genomes as a single database, and (II) the score computation step, which increases the score for samples with reads aligned across the genome compared to samples with reads aligned to a small section of the genome.

- Lines 79-80: Alignment is performed with SRPRISM as it guarantees the reporting of all equally good alignments (max 255) across all sequences in the database.

- Lines 159-161: The alignment and scoring were done as per the MetaNISH design described earlier. The scores for all 100 reference genomes for each SRA ID are reported by the pipeline.

- Lines 191-192: SRA reads were aligned to the set of reference genomes, and a score for each reference genome using padded coverage was obtained for SRA ids from January 2010 to November 2022, retrospectively.

A new figure was created to address the reviewer's concerns about multimappings (now the new Figure 1), and Lines 96-104 now reads: 

A scaling factor was applied to adjust the padding length for each read and genome based on the number of alignments, so this ensures that multiple mappings of a read to a genome do not exceed 100 Kb for each location. The score was the percentage of the genome covered by padded alignments. For example, read SRR11734778.40769.1 aligned to four C. auris genomes with only one location in each genome. Therefore, full padding of 100 Kb was used for each of the four alignments. However, read SRR11734778.1429600.1 aligned to four contigs on three genomes, as shown in Figure 1. Figures 1A and 1B show four alignments on two genomes, which reduced the padding to 25 Kb. Figure 1C shows two alignments on the third genome, which reduced the padding to 50 Kb. Padded coverage cannot extend beyond contig boundaries.

6. In Figure 1, the authors describe the pipeline used for the whole analysis and MetaNISH is just a small part of it. It would be important to clarify in the text and in the caption of the figure what were the pre-MetaNISH steps. It is not clear to me, for example, why the Docker logo is used without any reference to it in the manuscript. Moreover, as the “Curated Reference Database” is outside the MetaNISH, does this mean that MetaNISH can be run with any reference database created by the user? I think the manuscript would benefit if this Figure was substantially improved, clarifying not only the pre-MetaNISH steps, but also, and more importantly, the MetaNISH workflow and its input/output files.

Figure 2 (formerly Figure 1, addressed in the reviewer’s question) has been refined following the reviewer's suggestion, and more details have been added to make it self-explanatory. Additional text has been added to the caption of the figure to clarify the whole process. Docker was used to create a container to query data from the SRA db cloud, as indicated in Figure 2.

MetaNISH can be run with any reference database created by the user. This is stated in the manuscript as follows:

- Lines 125-130: The design of MetaNISH can be used for tracking any pathogen. It requires developing a reference set with representative genomes from all clades for the pathogen to be tracked and for nearby species. Doing so allows SRPRISM to find the best matches for each read among the genomes where a match can be expected. Then, empirical analysis is needed to find suitable parameters for padding and score threshold. If read properties change substantially over time, the alignment method and parameters may need revisited.

- Lines 327-328: Future efforts can adapt this tool to monitor other emerging pathogens and public health threats.

7. Regarding the benchmarking and comparison with KrakenUniq, what were the settings used for KrakenUniq? Also, how much time does MetaNISH require for the analysis of an SRA entry?

Settings used for KrakenUniq are stated in Lines 181-184: 

These scores were compared to KrakenUniq (24), a method for metagenomic classification that provides a quantitative measure of genome coverage. KrakenUniq was run with defaults, except no information was printed for unclassified sequences using parameter --only-classified-output.

Text describing requirements for MetaNISH running analysis is found at Lines 119-124:

The CPU time for the 4,000 runs used for determining the parameters varied from 25 seconds (for SRR7125652) to 17 hours 14 minutes (for SRR8550535) with a median time of 25 minutes. SRR7125652 has 86,554 paired reads, while SRR8550535 has over 418 million paired reads. We noted that SRPRISM, which takes almost all the time in the MetaNISH pipeline (as score calculation takes only a couple of seconds), can be run in multi-threaded mode with good scaling till eight threads, but we did not use that option for results reported here.

8. Sample SRR11734778 seems to have an outlier behavior when compared to the others (Table 1). Do the authors have an idea of what can be influencing the score in this sample?

Additional columns for padding added to Table 1 show that SRR11734778 is not an outlier, even with a padding of 10 Kb. Figure 1 also illustrates the analysis of a read from this read set.

9. Table 3 needs more explanation. Does “score” correspond to “genome coverage”? And the numbers correspond to the number of samples? Why are some numbers in red?

As stated in lines 98-99 of the MetaNISH Design section: 

The score was the percentage of the genome covered by padded alignments. 

So, the score corresponds to the percentage of padded genome coverage. The word ‘score’ has been added to the table title text referring to the above.

The numbers in the table correspond to the number of SRA runs (or samples); the word ‘samples’ has also been added to the table title text; the ones in red bold are samples whose metadata info is shown in Table 4; this information has been added to the table's footnotes.

Table 3, lines 249-253, has been updated:

Table 3. Binned padded genome coverage (score) of SRA runs (samples) with Candida auris hits.

 Score Ranges

Year [50-75) [75-85) [85-95) [95-100) 100*

2010 0 0 0 0 0

2011 0 0 0 0 0

2012 0 0 0 0 0

2013 0 0 0 0 0

2014 0 0 0 0 0

2015 0 0 0 0 0

2016 5 0 0 0 0

2017 0 0 0 0 0

2018 0 0 0 0 0

2019 1 0 2 4 6

2020 6 3 5 9 14

2021 0 0 0 0 0

2022 1 0 0 0 0

Total 13 3 7 13 20

Numbers in red bold correspond to the samples where C. auris was identified, and its metadata is shown in Table 4. 

*This is not an interval; it equals the number of runs with a score of 100. For all other columns, the interval is closed at the beginning and open at the end.

10. Regarding the MetaNISH implementation in GridRepublic, does it specifically screen for C. auris? Or for all the species present in the reference database?

Thank you.

While this implementation could be screening any of the reference genomes in the database, benchmark development is necessary to determine the appropriate set of reference genomes from all clades for any pathogen to be monitored; given this, we can only guarantee results for C. auris.

Reviewer #2: The authors describe an interesting system for searching for Candida auris using metagenomic data deposited in the NCBI public database. They present their analyses clearly and in detail, and also explain the many limitations of their method. This method looks very promising not only for C. auris, but also for searching for other fungal pathogens in the environment, determining the microbiome and proposing possible hypotheses for the emergence of C. auris.

One question; do you think these results can be modified if you used other strain from different clades to determine the parameters?

According to the stated in Lines 125-130. If the reference database comprises all the representative genome diversity known for a pathogen. In that case, a different strain is expected to not significantly modify the parameters to be determined unless it is a different species.

Reviewer #3: I find the manuscript very interesting and thorough, and the method they propose appears very sensitive in detecting C.auris infections. I have no major comments about the paper.

Minor comment: I find the description of "padding" confusing, and particularly this: "For each read and each genome, the padding length was scaled down by the number of alignments to the genome by that read. The score was the percentage of the genome covered by padded alignments." I suggest using a formula to make it clearer and more motivated, and possibly a figure to illustrate what the padding achieves.

A new figure was created to clarify the reviewer's concerns (now the new Figure 1), and new text has been added Lines 96-104 now reads: 

A scaling factor was applied to adjust the padding length for each read and genome based on the number of alignments, so this ensures that multiple mappings of a read to a genome do not exceed 100 Kb for each location. The score was the percentage of the genome covered by padded alignments. For example, read SRR11734778.40769.1 aligned to four C. auris genomes with only one location in each genome. Therefore, full padding of 100 Kb was used for each of the four alignments. However, read SRR11734778.1429600.1 aligned to four contigs on three genomes, as shown in Figure 1. Figures 1A and 1B show four alignments on two genomes, which reduced the padding to 25 Kb. Figure 1C shows two alignments on the third genome, which reduced the padding to 50 Kb. Padded coverage cannot extend beyond contig boundaries.

---

## [Decision Letter · Decision Letter 1]

5 Jan 2024

Finding Candida auris in public metagenomic repositories

PONE-D-23-27761R1

Dear Dr. Chow,

We’re pleased to inform you that your manuscript has been judged scientifically suitable for publication and will be formally accepted for publication once it meets all outstanding technical requirements.

Kind regards,

Ricardo Santos

Academic Editor

PLOS ONE

Additional Editor Comments (optional):

Reviewers' comments:

Reviewer's Responses to Questions

**Comments to the Author**

1. If the authors have adequately addressed your comments raised in a previous round of review and you feel that this manuscript is now acceptable for publication, you may indicate that here to bypass the “Comments to the Author” section, enter your conflict of interest statement in the “Confidential to Editor” section, and submit your "Accept" recommendation.

Reviewer #1: All comments have been addressed

Reviewer #3: All comments have been addressed

2. Is the manuscript technically sound, and do the data support the conclusions?

Reviewer #1: Yes

Reviewer #3: Yes

3. Has the statistical analysis been performed appropriately and rigorously? 

Reviewer #1: Yes

Reviewer #3: Yes

4. Have the authors made all data underlying the findings in their manuscript fully available?

Reviewer #1: Yes

Reviewer #3: Yes

5. Is the manuscript presented in an intelligible fashion and written in standard English?

Reviewer #1: Yes

Reviewer #3: Yes

6. Review Comments to the Author

Reviewer #1: I thank the authors for carefully addressing all my comments. Some minor comments on this new version are:

1. Figure 1 should be provided as a single figure with 3 panels, and not as 3 independent figures. The same applies to Figure 3.

2. Figure 2, Box in step 4 is out of format and species name in step 6 should be In italics

3. In Figure 2, I miss an arrow connecting step 1 and step 3? Or is the first arrow in step 2 in the wrong direction?

4. Table 3, numbers in red bold were clarified, but numbers in bold (not red) are not clarified.

Reviewer #3: I am satisfied with the responses of the authors to my and the other reviewer's comments. I have no further comments or questions about the paper.

7. PLOS authors have the option to publish the peer review history of their article (what does this mean?). If published, this will include your full peer review and any attached files.

Reviewer #1: No

Reviewer #3: **Yes: **Rahul Siddharthan

---

## [Editor Report · Acceptance letter]

11 Jan 2024

PONE-D-23-27761R1 

PLOS ONE

Dear Dr. Chow, 

I'm pleased to inform you that your manuscript has been deemed suitable for publication in PLOS ONE. Congratulations! Your manuscript is now being handed over to our production team.

Kind regards, 

on behalf of

Dr. Ricardo Santos 

Academic Editor

PLOS ONE